# A Lizardite–HCN Interaction Leading the Increasing of Molecular Complexity in an Alkaline Hydrothermal Scenario: Implications for Origin of Life Studies

**DOI:** 10.3390/life11070661

**Published:** 2021-07-06

**Authors:** Saúl A. Villafañe-Barajas, Marta Ruiz-Bermejo, Pedro Rayo-Pizarroso, Santos Gálvez-Martínez, Eva Mateo-Martí, María Colín-García

**Affiliations:** 1Posgrado en Ciencias de la Tierra, Universidad Nacional Autónoma de México, Ciudad Universitaria, Mexico City 04510, Mexico; aquarium@comunidad.unam.mx; 2Departamento de Evolución Molecular, Centro de Astrobiología (CSIC-INTA), Ctra, Torrejón-Ajalvir, km 4, Torrejón de Ardoz, 28850 Madrid, Spain; prayo@cab.inta-csic.es (P.R.-P.); sgalvez@cab.inta-csic.es (S.G.-M.); mateome@cab.inta-csic.es (E.M.-M.); 3Instituto de Geología, Universidad Nacional Autónoma de México, Ciudad Universitaria, Mexico City 04510, Mexico; mcolin@geologia.unam.mx

**Keywords:** hydrogen cyanide, alkaline hydrothermal environments, organic molecules, serpentine minerals, prebiotic chemistry

## Abstract

Hydrogen cyanide, HCN, is considered a fundamental molecule in chemical evolution. The named HCN polymers have been suggested as precursors of important bioorganics. Some novel researches have focused on the role of mineral surfaces in the hydrolysis and/or polymerization of cyanide species, but until now, their role has been unclear. Understanding the role of minerals in chemical evolution processes is crucial because minerals undoubtedly interacted with the organic molecules formed on the early Earth by different process. Therefore, we simulated the probable interactions between HCN and a serpentinite-hosted alkaline hydrothermal system. We studied the effect of serpentinite during the thermolysis of HCN at basic conditions (i.e., HCN 0.15 M, 50 h, 100 °C, pH > 10). The HCN-derived thermal polymer and supernatant formed after treatment were analyzed by several complementary analytical techniques. The results obtained suggest that: (I) the mineral surfaces can act as mediators in the mechanisms of organic molecule production such as the polymerization of HCN; (II) the thermal and physicochemical properties of the HCN polymer produced are affected by the presence of the mineral surface; and (III) serpentinite seems to inhibit the formation of bioorganic molecules compared with the control (without mineral).

## 1. Introduction 

Currently, it is undoubted that several cyanide species may have had a crucial role in the previous steps for the origin of life [1,2,3], either as an important source of precursors to building blocks of RNA, proteins, and lipids [4,5,6,7,8,9,10,11,12,13] or as important chemical intermediates in phosphorylation reactions [14]. These cyanide species may have been scattered throughout some primitive environments such as the hydrothermal systems, either submarine or subaerial [15,16,17,18,19,20]. Hydrothermal environments, both subaerial and submarine, are considered ideal systems that allowed chemical evolution on Earth [21,22,23]. It has been proposed that serpentine-hosted hydrothermal systems may support favorable conditions for prebiotic pathways due to the coexistence of different geochemical variables on them [21,24,25,26]. Serpentinites are rocks formed mostly of serpentine-group minerals derived from metamorphism of mafic–ultramafic rocks that were abundant during the Hadean–Archean eons [27,28,29,30]. In general, these hydrous magnesium silicates are formed after low-temperature ( <400 °C) hydration of ferromagnesian or magnesian minerals (e.g., olivine, orthopyroxene) formed in basic and ultrabasic rocks [31].

Because hydrothermal activity and the serpentinization process should have been common and widely distributed in the first 1000 Ma of the Earth’s history [24,26,29,32,33,34,35], some authors have highlighted the interactions among cyanide species and geochemical variables present in hydrothermal environments and their aftermath in origin of life scenarios [2,36,37,38,39,40,41]. 

Recent papers have shown that hydrothermal fluids, during the serpentinization process, can lead to the occurrence of carbonaceous matter on mineral surfaces [42,43,44]. In addition, it has been shown that alkaline conditions and the dynamic surroundings that are present in hydrothermal systems are crucial for the transformation of cyanide species (e.g., cyanide salts and hydrogen cyanide, HCN) into other organic precursors with interesting roles in pre-RNA world scenarios [11,12,45,46,47,48]. Therefore, it seems necessary to study the role of serpentinite during the polymerization of HCN in order to simulate a feasible primitive geochemical scenario such as the surroundings of an alkaline hydrothermal system. 

The polymerization of HCN has been widely studied [49,50,51,52,53,54,55]; however, few reports have focused on the polymerization of HCN onto mineral surfaces. Ferris et al. [56] found that the presence of montmorillonite inhibits the formation of oligomers because the clay decomposes the tetramer of HCN (i.e., diaminomaleonitrile, DAMN); this effect increases at higher temperatures [47]. However, Boclair and coworkers [57] reported that layered double hydroxides (LDH) can favor the self-addition of cyanide at alkaline pH. On the other hand, the ɣ-irradiation of an heterogeneous sample of HCN/Na-montmorillonite inhibited the amount of carboxylic acids formed [58]. In addition, it has been shown that the properties (e.g., structure, kind of deposition, morphology) of aminomalononitrile-based films are modified by the presence of surfaces (e.g., quartz, glass, and silica; [59]). There is no comprehensive information about the role of mineral surfaces during HCN polymerization. 

The dynamism of hydrothermal systems offers an interesting place for chemical reactions. For instance, the continuous transport of material along the surroundings of hydrothermal systems (i.e., the vent field that includes all active hydrothermal fluids (both at low (<100 °C) and high temperatures (<400 °C)) may involve the occurrence of thermolysis and polymerization reactions of raw material. Recently, we characterized a polymer formed from the thermolysis of HCN (i.e., HCN-DTP) [46] simulating a simple alkaline hydrothermal system. In this work, using the same synthesis conditions (i.e., HCN*_(l)_* 0.15 M, 50 h, 100 °C, pH > 10), we studied the role of serpentinite related to the physicochemical properties of the formed polymer (HCN-DTP/serpentinite) as well as the nature of the supernatant and the mineral coated by an organic layer (Figure 1). Finally, we discuss the implications for chemical evolution studies. 

## 2. Materials and Methods

### 2.1. Mineral/HCN Samples

Serpentinite was provided by Professor Fernando Ortega-Gutiérrez (Geology Institute, Universidad Nacional Autónoma de México, Ciudad Universitaria, 04510 Cd. Mx, Mexico). The sample was obtained from the Acatlán Complex, SW México [60]. In order to remove all the organic material on the sample, the following procedure was carried out: fragments of mineral (< 1 cm) were washed with a KOH solution (3% *v*/*v*) for 30 min (1 g mineral/10 mL solution). After that, the mineral was stirred in distilled water (30 min) to remove the KOH excess. Later, the sample was washed with HNO_3_ solution (3% *v*/*v*) (1 g mineral/10 mL solution) for 30 min. Finally, the mineral was cleaned with distilled water to remove the acid excess. Mineral was dried at room temperature. XRD analysis was performed for mineralogical characterization of the serpentinite sample. The lizardite polymorph predominated in the sample. XRD spectra (Figure 2) showed distinctive diffraction peaks corresponding to lizardite (91%), antigorite (5%), and minor traces of magnetite and brucite (≈ 4%). Lizardite has the structural formula M_3_T_2_O_5_(OH)_4_, where M is mainly Mg and T is Si, although several common elements can be present in Table 2. Al^3+,^ Ni, Mn^2+^, or Zn^2+^ [61,62,63]. In this mineral, 1:1 flat layers of sheets of SiO_4_ tetrahedra and sheets of MgO_2_(OH)_4_ octahedra are linked by hydrogen bonds. The most common polytypic is the stacking of three layers without any lateral shift [64].

To confirm that the mineral sample was not contaminated, a mass spectroscopy thermal analysis was carried out for the cleaned serpentinite sample. Peaks related to OH^−^ and H_2_O (at 619 and 696 °C) were detected; this corroborated that all organic material was removed from the serpentinite (figure not shown).

The HCN-DTP was synthesized in the presence of serpentinite as follows. HCN solution was produced in situ by the reaction between KCN and H_2_SO_4_ under argon atmosphere (for further details, please see Villafañe-Barajas et al., 2020 [46]). Once the desired concentration was reached (0.15 mol L^−1^), the pH of the HCN solution was adjusted (pH > 10) with KOH solution (0.1 mol L^−1^) to favor the availability of ^–^CN and the formation of HCN-polymers. Finally, aliquots of HCN solution (0.15 mol L^−1^, 5 mL) were prepared with 500 mg of the previously cleaned serpentinite in glass tubes and heated in a static system at 100 °C for 50 h. The selected temperature was consistent with the one found in the surroundings of alkaline hydrothermal environments. After treatment, three phases could be distinguished in the sample: I) supernatant (yellow soluble part), II) HCN-DTP (black polymer that was not adhered to mineral surface), and III) mineral + HCN-DTP (serpentinite covered by polymer). The three phases were analyzed by different analytical techniques (for more details, see Figure 1).

### 2.2. Analysis of Samples

#### 2.2.1. FT-IR Spectroscopy (FT-IR)

The spectra were collected with a FT-IR spectrometer (Nicolet Thermo Fisher ^®^, model Nexus 67, MA, USA, software OMNIC ) using CsI pellets. Using a DRIFT reflectance accessory (Harrik, model Praying Mantis DRP, New York, NY, USA), the spectra of Phases II and III in the 4000–450 cm^−1^ spectral region (spectral resolution of 2 cm^−1^) were obtained. 

#### 2.2.2. Thermal Analysis (TA)

A thermal analysis (thermogravimetry (TG), differential thermal analysis (DTG), and differential scanning calorimetry (DSC)) was performed with a TA instrument^®^ (SDTQ-600/Thermo Star). The method involved operating in isothermal mode (20 min) and a heating ramp of 10 °C min^−1^ until 1000 °C under inert atmosphere (argon, flux 100 mL min^−1^). The analysis of the main released species along dynamic thermal decomposition from fragmentation process, were carried out with a coupled TG–MS system using an electron-impact quadrupole mass-selective detector (model Thermostar QMS200 M3).

#### 2.2.3. XPS Spectroscopy Analysis

X-ray photoelectron spectroscopy analysis of Phase III was carried out in an ultrahigh-vacuum chamber equipped with a hemispherical electron analyzer and with the use of an Al K*α* X-ray source (1486.6 eV) with an aperture of 7 mm × 20 mm. The base pressure in the chamber was 3 × 10^−8^ mbar, and the experiments were performed at room temperature. The sample was analyzed by preparing a pellet containing a sample of approximately 100 mg obtained after grinding and pressing the Phase III (mineral + HCN-DTP). Phase III was also analyzed as raw sample (rock covered by the polymer). The peak analysis in different components was shaped, after background subtraction, as a convolution of Lorenztian and Gaussian curves. Binding energies were calibrated against the binding energy of the C 1s peak at 285.0 eV. Calculation of the atomic relationships between the identified elements was derived from integral peak intensities and sensitivity factors supplied by Handbook of XPS [65].

#### 2.2.4. Hydrolysis and GC–MS Analysis

To conduct a comparative analysis, a basic (NaOH 0.1N, 100 °C, 6 h) and acid (HCl 6N, 110 °C, 24 h) hydrolysis procedure was performed following previous reports [46,66]. After treatment, the samples (Phases I, II, and III) were analyzed by a GC system coupled to a 5975 VL MSD (Agilent^®^). The detection and characterization of different signals were performed as previously reported [9,46]. 

## 3. Results and Discussion

### 3.1. Fourier Transform Infrared (FT-IR) Spectroscopy 

The FT-IR spectra of Phases II and III were registered and compared with their respective control samples. That means that the Phase II was compared to the polymer control HCN-DTP synthesized in the absence of mineral and the Phase III to a control serpentinite sample (Figure 3). This was done as a first step to evaluate the effect of the serpentinite in the cyanide polymerization process. 

There were no appreciable differences between the spectra of the mineral alone and the Phase III (Figure 3A). The lack of differences between the naked mineral spectrum and the coated serpentinite is probably because the spectrum of the Phase III was registered using a pellet of the raw sample. As will be discussed below, the organic film represents a very low amount (in % weight) of the raw Phase III. Therefore, in relative proportion, the intensities of the mineral FT-IR features are much higher than those of the organic film; therefore, it was not possible to characterize the polymeric coating by this methodology. The FT-IR spectrum of the serpentinite described here (Figure 3A) is very similar to others spectra previously reported, with characteristic peaks centered at 3682 cm^−1^, related to MgO-H stretching vibration modes, and at 974 cm^−1^ with a shoulder at 1068 cm^−1^, corresponding to the Si–O–Si asymmetric stretching mode (for a detailed description of these FT-IR spectra, see, e.g., Rivero Crespo et al., 2019; [67]).

On the other hand, the FT-IR spectra of the Phase II and the polymer control HCN-DTP present the same main features (Figure 3B). Detailed interpretations of these spectra were presented in Villafañe-Barajas et al. [46] indicating the generation of a highly conjugated macrostructure dominated by oxygenated functional groups. However, some slight but significant differences can be observed between the Phase II and the polymer control HCN-DTP (inset plot in Figure 3B and the subtraction of the spectra in Figure 3C). The subtraction of the FT-IR spectra of the Phase II and the polymer control HCN-DTP (pink line in Figure 3C) presents a clear feature at 3682 cm^−1^ and additional features at 959 and 1067 cm^−1^, which can be related with the presence of residual serpentinite in the Phase II. The bands at 1803, 1675, and 1595 cm^−1^ may indicate a greater oxidation state for the Phase II, since these bands can be assigned to carbonyl compounds such as esters, ketones, amides, and carboxylic acids. Furthermore, the band at 1348 cm^−1^ can be related to -COO^-^ groups in carboxylic acids salts (in this case, the counterions can come from the residual serpentinite). The features centered at 2168, 2121, 2074, and 2043 cm^−1^ identified in the Phase II (inset plot Figure 3C) can be assigned to azide (-N=N=N), carbodiimide (-N=C=N), and isonitrile (N≡C) functional groups. Therefore, it seems that the serpentinite increases the hydrolysis in the HCN polymerization, resulting in highly oxidized products.

### 3.2. Thermal Analysis

Thermal analysis allowed characterization of the thermal behavior of the samples and finding notable differences between HCN polymers that were not evident using FT-IR spectroscopy. DTG and DSC curves are considered fingerprints to characterize and distinguish among HCN polymers with very similar FT-IR spectra [68,69]. As was done in the previous sections with the FT-IR spectra, the thermal curves of the Phase II were compared with those of the polymer control HCN-DTP and the curves for the Phase III with the control clean serpentinite (Figure 4). In addition, the thermal analysis of the Phase III was carried out under an air atmosphere. In accordance with previous reports, the thermogravimetric behavior of the samples was divided into three stages: I) drying stage (<150 °C), II) pyrolysis stage (150–450 °C), and III) carbonization stage (>450 °C) [46,68,69,70,71,72]. The total weight loss of serpentinite was 12.6%, which agrees with previous reports, together with the DTG doublet in the carbonization stage (Figure 4A,C; Table 1) and with the DTA sharp exothermic peak at 823 °C [73]. The thermal decomposition of this mineral leads to the dehydroxylation of the structure; the bound hydroxyl groups are removed from the serpentinite and liberated as water vapor [74]. The release of these groups reaches its maximum peaks at ~619 and ~696 °C as shown in the DTG curve (Figure 4C; Table 1). Likewise, the DSC and DTA curves (Figure 4E and Figure 5) show a sharp exotherm peak at ~823 °C, which indicates the complete formation of olivine (i.e., forsterite; Mg_2_SiO_4_) ([74,75].

As with the FT-IR analysis, the thermal analysis does not show appreciable differences between the Phase III and serpentinite (Figure 4A,C,E and Figure 5; Table 1). There was a mass loss of only ~2% during the last thermal stage (> 450 °C) (Figure 4A). This suggests that there was not a significant amount of polymer covering the mineral surface, and, in consequence, the thermal profile obtained was dominated by the thermal behavior of serpentinite. However, for the Phase III, the total weight loss was 10.8%, an amount lower than the weight loss for the control sample, the clean serpentinite. Since dehydration is the unique thermal degradation processes for the serpentinite, it seems that the polymeric coating might partially prevent the release of water. This also would influence the phase change, because a reduction in the exothermicity of the forsterite formation was observed (Figure 4E and Figure 5). In addition, no thermal degradation of the coating was observed in the thermal curves herein considered. The next section discusses that this thermal decomposition seems to be minor. Furthermore, the thermal degradation of the Phase III in the presence of air led to an additional weight loss of 0.6 % compared to the weight loss obtained by heating under Ar atmosphere (Figure 4A). This result can be related to the total degradation of the polymeric coating, indicating a very low organic contribution (relative percentage in weight) from the film covering the mineral. 

On the other hand, the presence of air did not affect the thermal behavior of the serpentinite, since there are not great differences between the thermal curves recorded in the presence of air (previously reported) with those recorded using an inert atmosphere of Ar. Nevertheless, the changes observed in the shapes of the DSC and DTA curves of the Phase III registered in the presence of air seem to be due to the polymeric coating, although without highly significant variations with respect to the clean control serpentinite. These results might indicate a physical (physisorption) interaction between the polymeric coating and the mineral surface instead of a strong chemical interaction (chemisorption). This fact is important to elucidate the possible role of the mineral in the full process of the cyanide polymerization and to understand the interaction between the mineral and the organic polymer. To the best our knowledge, this is the first time that the thermal analysis of a covered mineral by a HCN-derived polymer has been described.

In regard to the Phase II, a mass loss of around 10 wt% was observed during the drying stage (<150 °C), which is a consequence of the release of volatile compounds retained in the polymer, predominantly H_2_O. This value is consistent with previous reports [46,68,69,72]. In the second stage, the pyrolysis stage (150–450 °C), the mass loss was also 10 wt%. This value is considerably lower than those previously reported (approx. 25 wt%) [46,68,69,72]. This suggests that, as the second thermal stage has been associated with the decomposition of the side groups on the main chain, the polymer has no highly stable side structures. Two clear DTG peaks appear at 174 and 290 °C. Similar peaks also appear in the DTG curve for HCN-DTP (Figure 4B) [46]. In addition, a shoulder around ~135 °C is detected. The third step, the carbonization stage (>450 °C), showed the most significant differences between the Phase II and the HCN-DTP previously synthesized [46]. The sample yield was approximately ~63 wt% of char residues at the end of the ramp temperature. This is unexpectedly high, which is because of the fact that HCN-DTP yields around 15 wt%, while other HCN-derived polymers have 20 wt% [46,68,69,72]. In addition, this value represents almost the double percentage of the original weight reported (i.e., 36.4%) for the black polymer formed by the thermal decomposition of formamide [76]. It seems, as is the case for the coating film (Phase III), that this Phase II is highly thermally stable. Three signals are evident in the DTG curve: first a slight shoulder around 550 °C and then two predominant peaks at 654 and 691 °C. These peaks match with the signals detected in HCN-derived polymers synthesized from NH_4_CN and DAMN [72]; however, this does not mean that identical structures are present. For instance, although there is no clear signal after 800 °C, as in other experiments performed at high temperatures (>80 °C) and high concentrations (>0.1 M) [46,72], there is a change in slope that starts at ~800 °C in the sample synthesized in presence of the mineral (Phase II). Finally, the DSC curve of the Phase II shows only three clear endothermic events at 71, 164, and 649 °C (Table 1). These signals correspond to evaporation of the absorbed water and the two main decomposition processes of the polymer [46,72]. Although the FT-IR spectra of the Phase II and the HCN-DTP previously synthesized are essentially alike, the thermal analysis shows important specific thermal fingerprints for each sample. The main difference is shown in the comparison of the DTG curve from Phase II and the HCN-DTP [46], where there is a peak after 900 °C in the control sample. Likewise, the Phase II shows a higher release of volatile species (e.g., H_2_O) in the first thermal step (i.e., <70 °C). This considerable difference might be associated with an increment of hydroxyl groups in the polymer as a consequence of interactions with the serpentinite structure. Also, the high amount of char for the Phase II could be related with a higher degree of oxidation, in agreement with the FT-IR spectra. It has been proposed that an increase in the thermal stability of the HCN-derived polymers is related to a higher content of oxidizing groups in the macrostructures, which eventually leads to an increase in cross-linking [69]. On the other hand, the residual amount of serpentinite in the Phase II may be very small, because its characteristic thermal peaks are not identified in any of the thermal curves (TGA, DTG, or DSC) of the Phase II (Figure 4B,D,F).

### 3.3. Mass Spectroscopy Thermal Analysis

Serpentinite control, Phase II, and Phase III were analyzed by in situ mass spectrometry (Figure 6) to gain more information about the chemical species released in each thermal step shown in the DTG curves. We considered the same three thermal stages to compare the thermal behavior. The serpentinite control shows only signals associated to OH^−^ (*m/z* = 17) and H_2_O (*m/z* = 18) at 621 and 700 °C, which signals are consistent with the predominant peaks related to the release of hydroxyl groups from serpentinite as is confirmed by the DTG curve (figure not shown) and is in agreement with previous results [73]. Comparatively, the Phase III shows four clear signals. Three of them coincide around ~620 and ~700 °C and could be associated with NH_2_ (*m/z* = 16); OH^−^, NH_3_ (*m/z* = 17); and H_2_O, NH_4_^+^ (*m/z* = 18). In addition, an appreciable signal at 214 °C is related to HCN (*m/z* = 27) (Figure 6A). Since the serpentinite control does not show peaks below 600 °C, the peak at 214 °C can be directly related to the thermal decomposition of the organic coating (Figure 6A). Considering the intensity of these TG–MS data, the amount of coating on the serpentinite is small, though it seems to partially protect the mineral of dehydration. Further works are needed to determine the thickness and the nature of these films synthesized under hydrothermal conditions. Only one previous paper has reported the kinetics of the deposition of AMN-derived polymers, but it did so at room temperature [59] and did not focus on the thermal stability of this new series of coatings.

Phase II shows several changes related to the signals observed in the HCN-DTP control sample [46]. Figure 6B displays the predominant signals linked to NH_2_ (*m/z* = 16; T = 659 °C); OH^−^, NH_3_ (*m/z* = 17; T = 67, 171, and 662 °C); H_2_O, NH_4_^+^ (*m/z* = 18; T = 67, 241, 268, 557, and 685 °C); and NO (*m/z* = 30; T = 155 and 668 °C). Though these species and peaks were also present in the HCN-DTP, the Phase II has new, well-defined signals associated with NH_2_, OH^−^/NH_3_, and NO around ~660 °C. In addition, the *m/z* = 30 profile does not have the same broad peak around ~160 °C as the control sample (Figure 6B). Two clear peaks are shown for the profiles associated with N/CH_2_ (*m/z* = 14); CO, N_2_ (*m/z* = 28); and N_2_H/HCO (*m/z* = 29) at ~650 and 750 °C (Figure 6C). These signals contribute to the broad peak shown in the DTG curve at the third stage (>450 °C). The thermal profile is totally different, even though these species are released in the HCN-DTP. First, the ion current is considerably higher, and it is thus impossible to identify peaks before 600 °C as is possible in the control sample (e.g., 172 and 283 °C). Moreover, the Phase II does not present peaks after 900 °C as is evident in the HCN-DTP. 

In general, the signals in the pyrolysis (150–450 °C) and carbonization stages (>450 °C) are considerably different to the thermal profile of the polymer synthesized in the absence of mineral. The main differences are the predominant peaks around ~660 °C and the absence of peaks after ~900 °C (Figure 6D,E,F). The peaks around ~660 °C are the result of the contribution of several carbon and/or nitrogen species, e.g., C^+^ (*m/z* = 12), NH (*m/z* = 15), NCO (*m/z* = 42), HNCO/HOCN (*m/z* = 43), CO_2_/HC(=NH)NH_2_ (*m/z* = 44), and HCONH_2_ (*m/z* = 45), which suggest the dominance of decarboxylation and/or deamination mechanisms. Other peaks displayed in the same species profiles below 300 °C (e.g., 130, 180, 255, 282) are present in the polymer synthesized without mineral. However, the peaks at 392, 500, and 550 °C are only present for the CH, NH, NCO, CO_2_/HC(=NH)NH_2_, and HCONH_2_ species in the Phase II (Figure 6E,F). New signals without clear assignments are present in the polymer synthesized in the presence of mineral (e.g., *m/z* = 13 and *m/z* =46). Interesting, the profiles associated to cyanide species, CN^-^ (*m/z* = 26) and HCN (*m/z* = 27), are totally different from the control HCN-DTP sample. For instance, the profile of the sample without mineral shows a broad peak that starts at 423 °C and disappears after 900 °C. Instead, in the presence of serpentinite, cyanide species profiles only show three peaks between 640 and 780 °C (Figure 6F). These profiles resemble the general pattern of the DTG curve from the Phase II. Hence, decyanogenation of the polymer takes place mainly at 650 °C. In general, the thermal profile of Phase II shows that decarboxylation, deamination, and/or decyanogenation mechanisms occur mainly around ~660 °C. The absence of this thermal step and the predominant peak after 900 °C in the polymer synthesized without mineral suggest that the presence of serpentinite considerably affects the thermal and structural properties of the HCN-derived thermal polymers. In addition, the possible decyanogenation and degradation process of the Phase III is centered at 214 °C, suggesting a higher thermal lability of the coating than the insoluble black solid, Phase II; this implies that the coating and the insoluble black solid have different structural natures.

Table 2 shows a review of each volatile species contribution related to the thermal stages previously described. In summary, the peak at 124 °C and the two signals after 900 °C do not appear on the DTG curve of the polymer synthesized in the presence of serpentinite. In addition, the serpentinite contributes with an important amount of hydroxyl groups released in the first thermal step (i.e., 62 °C). Finally, the peak around 550 °C is unique in this sample.

### 3.4. XPS Analysis

The general thermal analysis suggested important physicochemical differences between the HCN-DTP synthesized with and without serpentinite, so X-ray photoelectron spectroscopy analysis was carried out to elucidate the interactions between the organic coating and the mineral surface. Figure 7A shows the photoemission spectra of both the clean serpentinite and the Phase III. The data were registered by using pellets from both samples. The serpentinite control shows clear signals related to its atomic composition (Mg, Si, and O) in strong agreement with previous reports [77]. In this case, the spectra of the clean serpentinite and the Phase III do not show significant chemical changes (Figure 7A). This could be a result of the low amount of organic polymer deposited onto the mineral surface and diluted in the bulk of the pellet, which may have been insufficient to obtain characteristic or specific signals from the organic film. Some previous studies by our group, where the characterization HCN-derived polymers and *tholins* was performed, reported that ~25% in weight of organic material by total mass of sample (mineral + organic) is necessary [71,78,79] to identify the chemical features of that material without doubt. 

However, when the XPS spectrum of the Phase III was performed using a coated serpentinite grain, the differences were remarkable (Figure 7B). The main difference among the samples containing an organic coating on the surface is that the mineral signals are attenuated. This can be explained by one main reason: the XPS technique is mainly sensible to the first surface layers of the sample; therefore, once the polymer covers the mineral surface forming a film, the mineral surface stays hidden underneath the organic material. Because of this phenomenon, a signal assigned to N 1s, the fingerprint of the polymeric film, is observed in the Phase III (inset plot in Figure 7B,C). The first component (at 401.8 eV) can be assigned to ammonium cations, and the second one (at 403.3 eV) to azide groups (R-N=N=N-) or nitrites (Figure 7B). The assignation of these components is different to those of others coatings obtained from AMN at room temperature (although the N 1s signal is significant, Thissen et al. performed deep analysis of C 1s spectra and the binding energies were referenced to aliphatic hydrocarbons peak at 285.0 eV [80] or insoluble cyanide polymers (397.6 eV to imine and/or heterocyclic groups and 398.7 eV to amides [71]. This result is consistent with the fact that the structural nature of the HCN polymers is directly dependent on the experimental synthetic conditions, as well as the chosen monomers used in their syntheses [72]. In the present case, the nature of the mineral substrate could also be considered to explain these differences. Recent works have suggested that the presence of inorganic surfaces (i.e., silica) can affect the properties (e.g., morphology and change of composition film) of AMN-based films [59]. Likewise, Ball [81] reported that AMN-based films deposited on amorphous carbon can affect its electrochemical activity as a consequence of the change in the composition, morphology, and thickness of the film. In addition, he suggested that long deposition times (21 h) produce thicker films. Because our experiments were performing using 50 h of reaction time at a higher temperature, the attenuated XPS signals from the mineral (Figure 7B) may be the consequence of a thick coating that covers the mineral surface. Nevertheless, we cannot discard molecular interactions and/or chemical reactions catalyzed by the mineral between the HCN-DTP and serpentinite once the polymer was formed (e.g., effects of the hydroxyl groups and/or Fe/Mg atoms during polymerization reaction). Additional systematic experiments are needed to understand the real interactions between the HCN polymers and inorganic surfaces. As a result, it seems that there is a physical adsorption of the polymeric coating on the serpentinite mineral, but chemisorption cannot be ruled out. Under the point of view of origin of life studies, these considerations about coating mineral superficies are of interest because they have not been previously taken into account and may increase the chemical space of prebiotic chemistry due to the potential redox properties of these coatings.

### 3.5. GC–MS Analysis of Hydrolyzed Samples

The dynamism of hydrothermal systems offers important temperature (25–400 °C) and pH gradients (pH = 2–11) along the hydrothermal fields [29,46,82,83,84]. Hence, the polymers formed in these environments are probably continuously exposed to thermolysis and hydrolysis reactions. Because hydrolysis conditions (i.e., heating time and pH value) are directly related to the amount of organic molecules released [7,66,85], we carried out a hydrolysis procedure (both alkaline and acid hydrolysis at 100–110 °C) for each phase to identify some polar organic compounds associated with each sample. In addition, we compared the detected molecules with those found in the HCN-DTP synthesized in the control experiment (in the absence of serpentine). Figure 8 summarizes all the organic compounds identified in this work.

The predominant species, both in acidic and basic hydrolysis, are glycerol, glycolic acid, succinic acid, orotic acid, stearic acid, and palmitic acid (Figure 9). However, some trends can be distinguished among phases and hydrolysis conditions. For instance, after acidic hydrolysis, carboxylic acids predominated, such as lactic, glycolic, and succinic acid (Figure 9A,C) and only after acid hydrolysis was 2, 5-dihydroxy pyrazine was present. Interestingly, some fatty acids were detected after the basic hydrolysis of the Phase I and II (Figure 9B,D). Related to this, Takahashi and coworkers [86] described the synthesis of fatty acids from HCN using organic solvents and high temperatures (>100 °C). Likewise, Eschenmoser [87] proposed a hypothetical pathway based on the hydrolysis of diaminomaleonitrile (DAMN) and other intermediates previously detected [46] to explain the formation of fatty acids.

In basic hydrolysis there was a broader spectrum of hydroxylated species in addition to the species detected by the acid hydrolysis, such as α-hydroxybutiric acid, oxalic acid, malonic acid, hydroxymalonic acid, aminomalonic acid, and other fatty acids (heptanoic acid, decanoic acid, and dodecanoic acid). Carbamate, glycine, orotic acid, and urea were also detected. Notably, in all cases, glycerol was present. 

Regarding the Phase III, even using a higher amount of sample (50 mg), there were no clear signals associated with important organic compounds. As mentioned before, this could be a result of the low amount of polymer deposited onto mineral. Only carbamate and glycerol were identified after basic hydrolysis. Thissen et al. [88], in analogous deposition experiments performed on gold substrates, proposed that hydrolysis of imine groups promotes the formation of carbonyl compounds. In our case, the hydrolysis reactions could also lead to the formation of these compounds. Some uncertain signals, probably attributed to alkanes, could be distinguished after acid hydrolysis. 

Minerals could affect the nature of the formed product, even if the exact mechanisms for this are not fully understood. For example, González-López and coworkers [89] found that olivine biases the products formed by thermolysis of acetic acid and formic acid. In our experiments, the presence of serpentinite had interesting repercussions in the diversity of organic molecules detected after hydrolysis treatment (Figure 8 and Figure 9). On the one hand, after acid hydrolysis in presence of mineral, some carboxylic acids, fatty acids, and pyrazine in the Phase I and Phase II were identified. The presence of mineral seems to inhibit the formation of N-heterocyclic compounds, as is evident in the control experiment (HCN-DTP). On the other hand, after basic hydrolysis of samples synthesized in the presence of inorganic surfaces, some amino acids and predominantly fatty acids were released from Phase I and II. Orotic acid was the only cyclic compound detected after both hydrolysis treatments for the samples synthesized in the presence of mineral. 

## 4. Outlook on Studies about Prebiotic Molecular Complexity

The current knowledge about the formation of HCN-derived polymers shows that their features depend on synthesis conditions, including concentration, temperature, presence of oxygen, time of reaction, and raw material [11,46,48,53,54,55,69,72]. Alkaline hydrothermal systems have been pointed out as very versatile and crucial environments for chemical evolution and, eventually, for origin of life scenarios [38,39,40,45,90,91,92,93,94,95]. Considering the new perspectives of prebiotic chemistry (that suggest to take into account the dynamism of environments, as well as the interactions among their geochemical variables; [29,46,96]), we tested the role of serpentinite during the polymerization of HCN as a first approximation of a simple simulation of an alkaline hydrothermal system. The results suggest that the presence of this mineral affects the thermal properties of HCN-DTP. For instance, the release of volatile species as a function of temperature increment having a maximum peak around ~660 °C is considerably different from the control sample. In addition, some interesting carbon and/or nitrogen volatile species are associated with specific thermal events. 

Although the synthesis of organic compounds from serpentinization fluids in ultramafic systems has been reported [97,98], our results suggest that the presence of serpentinite could affect the chemical structure of the formed polymer (in this case HCN-DTP). Hence, it could decrease the diversity of organic molecules released after hydrolysis treatment. Markedly, this inorganic surface inhibited the formation of polar organic compounds and mainly N-heterocyclic compounds, unlike the control experiment (HCN-DTP) [46]. These results raise interesting questions: Is molecular complexity, considering the polymerization of HCN as a prebiotic pathway, reduced in mineral-rich environments? How can the addition of more geochemical variables affect the reactivity of the system? What is the potentiality of these coatings? 

Considering the role of minerals in HCN polymerization, previous reports have suggested that montmorillonite inhibits the formation of oligomers [47,56] and reduces the amount of carboxylic acids formed [58]. However, this does not mean that all mineral surfaces may have the same effect. Related to the second question, an interesting study showed that the addition of salts during the HCN polymerization process reduced the diversity of organic molecules, and ammonia had the opposite effect [9]. Compared with similar experiments that simulated alkaline hydrothermal environments using NH_4_CN as initial reagent and microwave conditions as energy source, the polar organic compounds identified by the same GC–MS method were considerably lower in our case [48].

On the other hand, the considerable thermal stability of HCN-DTP suggests that its persistence under hydrothermal conditions as well as the presence of coated mineral surfaces could establish new steps in chemical evolution, e.g., acting as semiconductors and catalysts, which are phenomena with important repercussions in the development of new chemical pathways [72] such as photocatalyst reactions [99]. In this way, understanding the properties and roles of these complex materials can shed light about specific catalytic reactions and the formation of some constituents of primitive chemical cycles that were necessary in the first steps of the origin of life [100]. 

These results indicate that the contribution of different geochemical variables in the same experiment, which tried to simulate the dynamism in these primitive environments, could modify the production of organic compounds. This does not mean a “knock-out” for these systems and their role in chemical evolution. For example, after basic hydrolysis of samples synthesized in presence of serpentinite, the release of some fatty acids is possible. The availability of these molecules and their eventual concentration in microporous matrices of alkaline vents can be enough to precipitate into vesicles [101]. Likewise, depending of pH conditions and salt concentrations, some fatty acid membranes resist extreme environments [102]. On the other hand, even though purines were not synthesized in the presence of mineral, orotic acid was released after hydrolysis treatment (acid and basic). Recently, the importance of orotic acid as a starting point in chemical pathways to RNA has been pointed out [103,104,105]. Orotic acid can form metal complexes with common ions in hydrothermal conditions (e.g., Cu^2+^, Mn^2+^, Zn^2+^) [106], which complexes could establish new chemical interactions with other organic molecules or surfaces. 

## 5. Conclusions

New perspectives in prebiotic chemistry suggest the necessity of considering the dynamism of primitive environments. Since minerals and organic molecules should have interacted continuously during early Earth, important phenomena could have developed. Common environments like alkaline hydrothermal systems could have important repercussions in the synthesis and evolution of complex materials such as HCN-DTP and coatings of mineral surfaces. The results suggest that the presence of serpentinite affects the thermal properties of the formed polymer as well the carbon and/or nitrogen volatile species released in specific thermal events. After hydrolysis treatment, several organic molecules with interesting importance in pre-RNA scenarios were identified. Although the effect of mineral surfaces in chemical evolution process has been widely investigated, the focus on coating mineral surfaces is recent. The effect of the presence of mineral surfaces in polymerization reactions and their repercussions in the physicochemical nature of the polymers formed is a new area that introduces a novel vision in prebiotic chemistry.

## Figures and Tables

**Figure 1 life-11-00661-f001:**
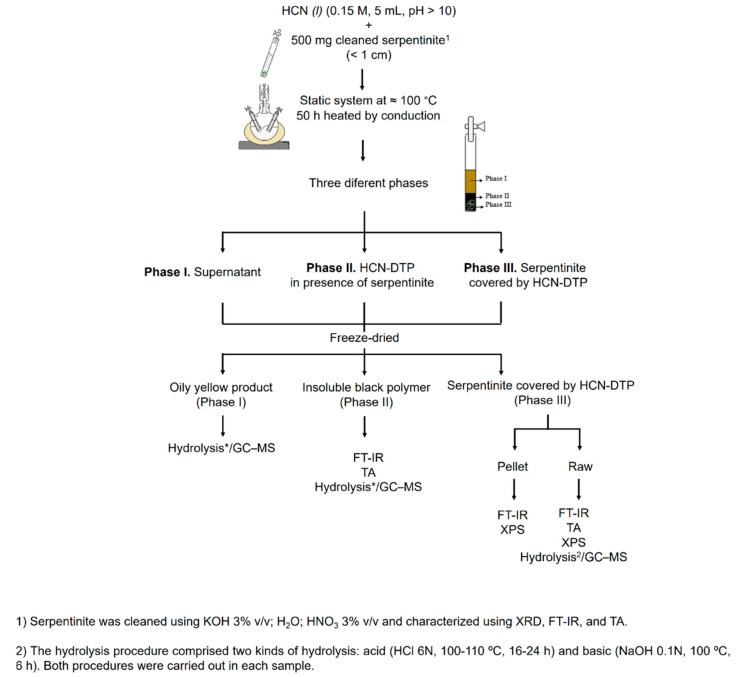
Processes followed for the synthesis and characterization of the HCN-DTP in presence of serpentinite.

**Figure 2 life-11-00661-f002:**
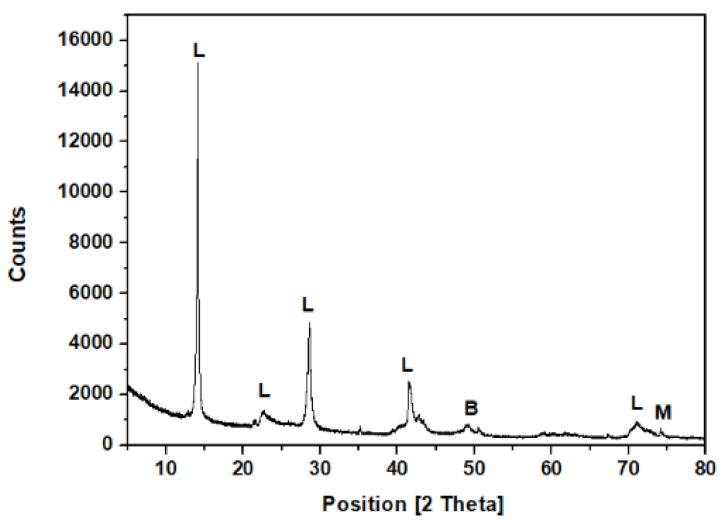
X-ray pattern for the serpentinite rock sample. Diffraction peaks were assigned based on previous works [61,63]. Legend: (L) lizardite Mg_3_Si_2_O_5_ (OH)_4_, (M) magnetite (Fe_3_O_4_), and (B) brucite (Mg(OH)_2_).

**Figure 3 life-11-00661-f003:**
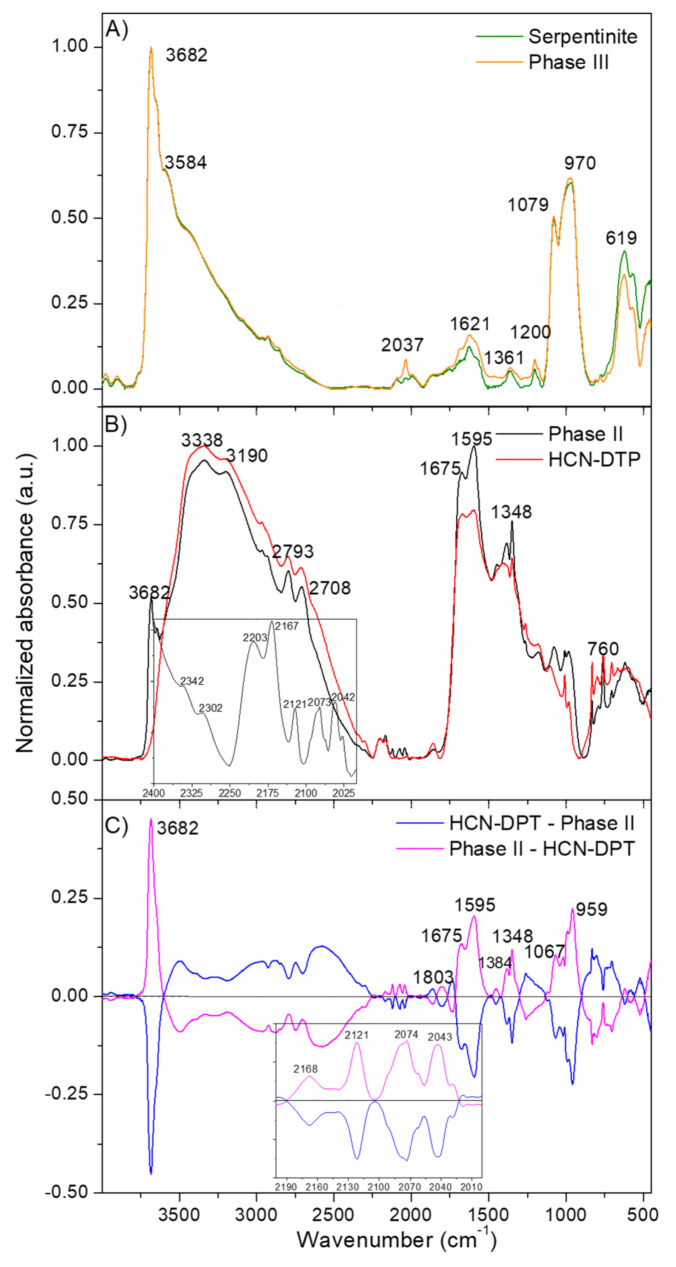
FT-IR spectra of (**A**) net serpentinite (clean control mineral) and serpentinite coated in a HCN-derived polymeric film (Phase III); (**B**) HCN-DTP polymer synthesized in the presence of serpentinite (Phase II) and control HCN-DTP polymer (synthesized in the absence of serpentinite); and (**C**) subtraction of the FT-IR spectra of the Phase II and the control HCN-DTP polymer.

**Figure 4 life-11-00661-f004:**
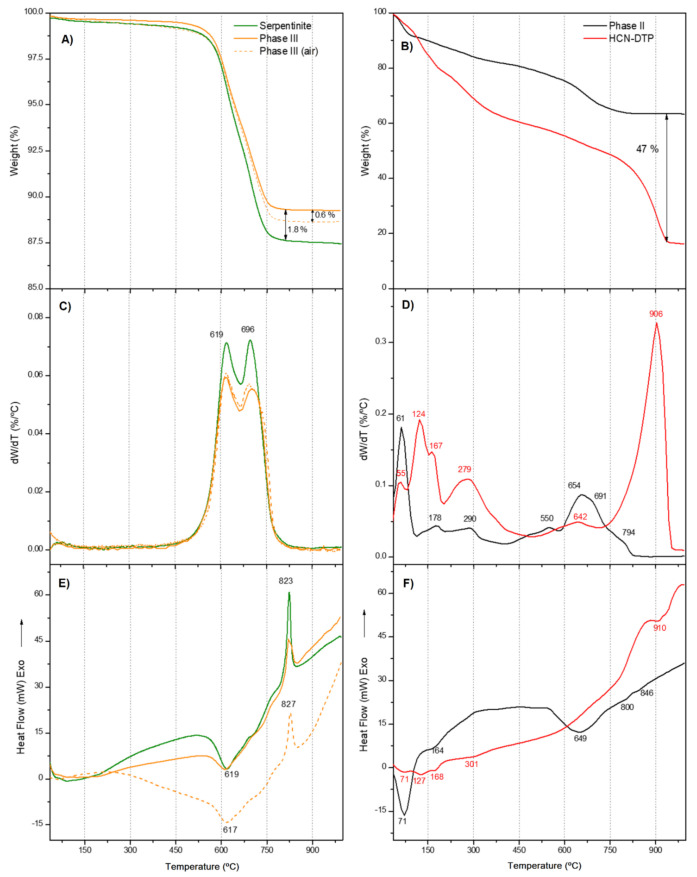
(**A**) and (**B**) TG curves; (**C**) and (**D**) DTG curves; (**E**) and (**F**) DSC curves. Note that an important amount of residue remained even at high temperatures. The most important decomposition step occurred along the third thermal stage. The identified stages were (I) drying stage (<150 °C), (II) pyrolysis stage (150–450 °C), and (III) carbonization stage (>450 °C).

**Figure 5 life-11-00661-f005:**
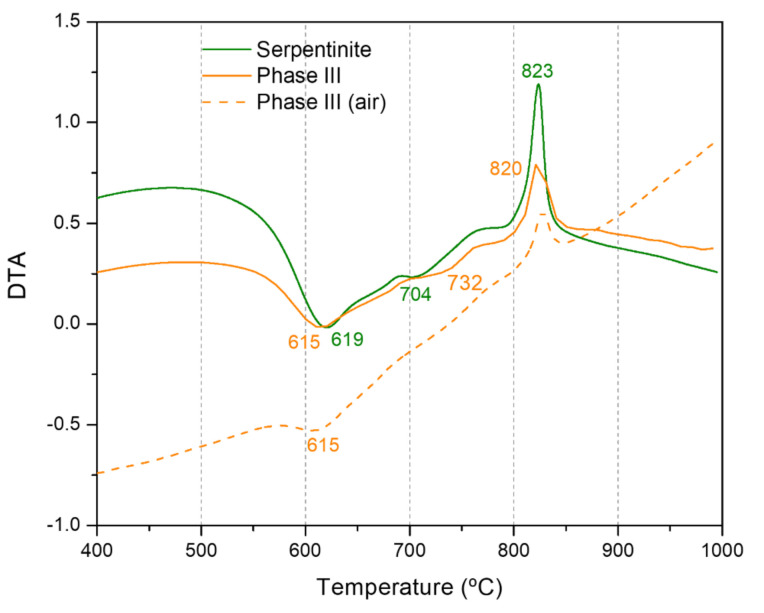
DTA analysis of serpentinite, phase III and phase III (air).

**Figure 6 life-11-00661-f006:**
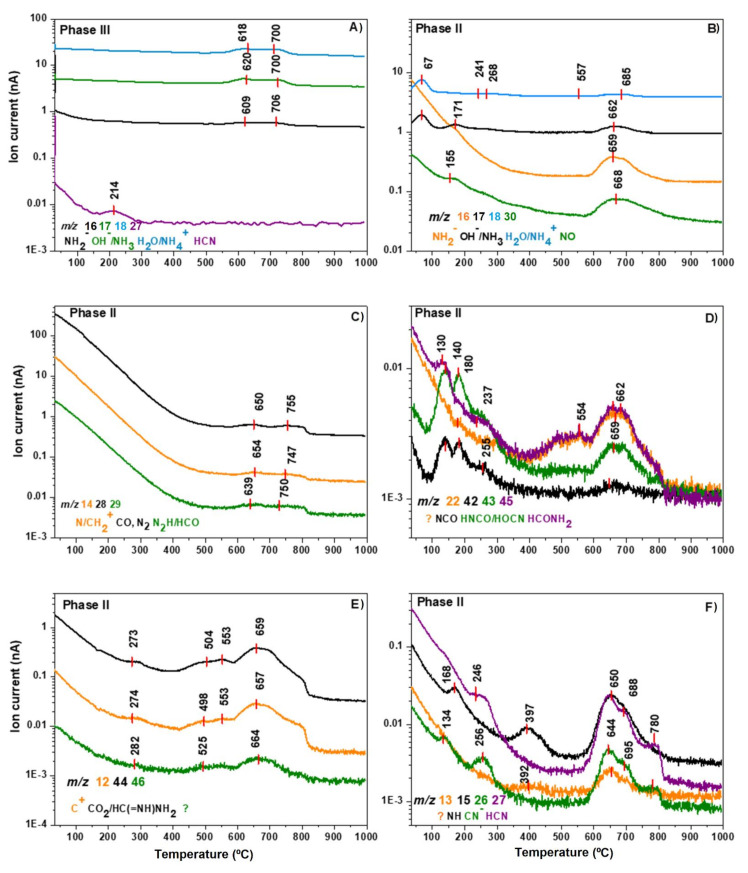
Ion intensity curves: (**A**) for Phase III; (**B**–**F**) for the Phase II. All signals are indicated in Table 2 with their correspondent DTG peaks. The predominant peaks on the third stage are the result of the contribution of several carbon and/or nitrogen species.

**Figure 7 life-11-00661-f007:**
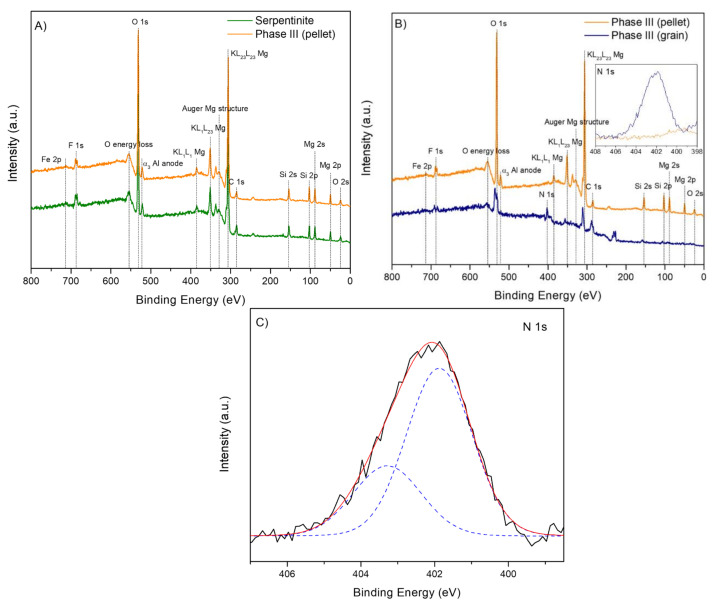
Photoemission spectra of samples. (**A**) XPS overview photoemission spectra of serpentinite control and Phase III (pellet); (**B**) XPS overview photoemission spectra of Phase III (pellet and grain); and (**C**) XPS spectra of N 1s core level peak of Phase III (grain sample), recognized as the fingerprint of the polymeric film.

**Figure 8 life-11-00661-f008:**
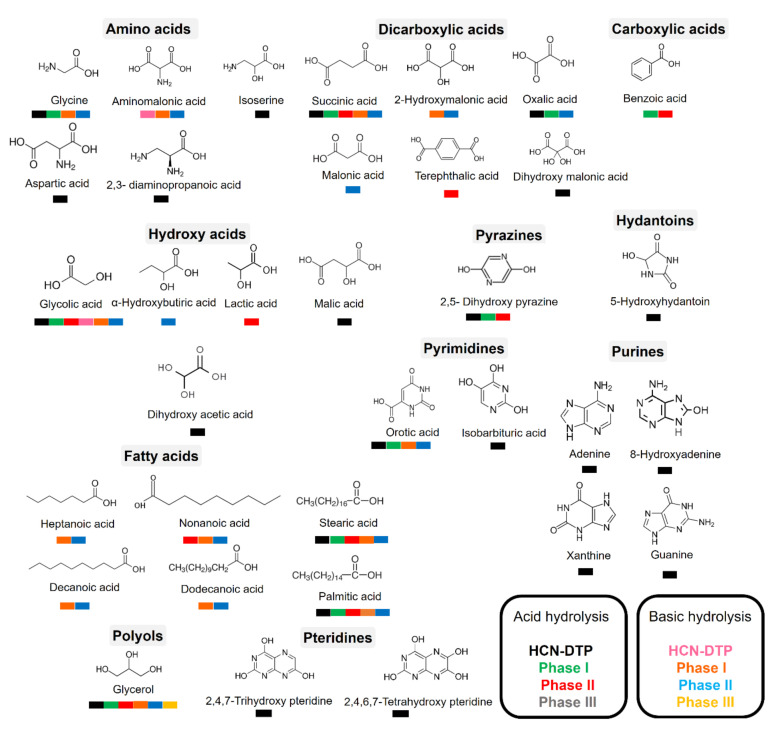
Organic compounds identified in samples after hydrolysis treatments. The colors indicate the sample from which the organic compounds were released.

**Figure 9 life-11-00661-f009:**
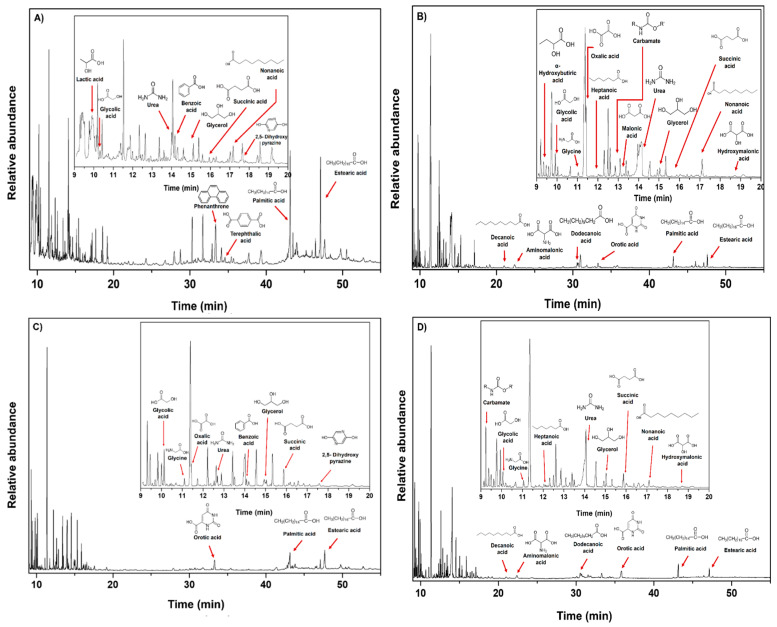
Organic compounds identified in samples after hydrolysis treatments: (**A**) acid hydrolysis for Phase II; (**B**) basic hydrolysis for Phase II; (**C**) acid hydrolysis for Phase I; and (**D**) basic hydrolysis for Phase I. The predominant species, both for acidic and basic hydrolysis, were glycerol, glycolic acid, succinic acid, orotic acid, stearic acid, and palmitic acid.

**Table 1 life-11-00661-t001:** Characteristic temperatures for the thermal decomposition of samples. DTG maxima with the corresponding rates of weight loss, dW/dt, and DSC peaks observed in the samples.

	Stage I (25–150 °C) Evaporation			Stage II (150–450 °C) Low Thermal Decomposition			Stage III (450–1000 °C) High Thermal Decomposition		
	DTG		DSC	DTG		DSC	DTG		DSC
Sample	T_max_ (°C)	dW/dt (wt%/°C)	T_peak_ (°C)	T_max_ (°C)	dW/dt (wt%/°C)	T_peak_ (°C)	T_max_ (°C)	dW/dt (wt%/°C)	T_max_ (°C)
Serpentinite							619	0.07	619
							696	0.07	823
Phase II	61	0.18	71	178	0.04	164	550	0.04	649
				290	0.04		654	0.08	
							691	0.07	800
							794	0.03	846
Phase III							619	0.06	619
							696	0.05	823
Phase III (Air)							619	0.06	617
							696	0.05	827
HCN-DTP	55	0.11	71	167	0.15	168	642	0.05	636
	124	0.20	127	279	0.11		906	0.34	910
				288	0.11	301	921	0.27	938

**Table 2 life-11-00661-t002:** Summary of detected volatile species in Phase II. The MS peaks are associated with each thermal stage. Orange = Stage I. Evaporation (25–150 °C); Blue = Stage II. Low thermal decomposition (150–450 °C); Green = Stage III. High thermal decomposition (450–1000 °C).

Probable Species	MS Peaks (*m/z*)	TG-MS Peaks for Phase II				
		62	174	285	550	654
C+	12					
?	13					
N,CH_2_^+^	14					
NH	15					
NH_2_	16					
OH^−^/NH_3_	17					
H_2_O/NH_4_+	18					
?	22					
-CN	26					
HCN	27					
CO,N_2_	28					
N_2_H,HCO	29					
NO	30					
NCO	42					
HNCO/HOCN	43					
CO_2_/ HC(=NH)NH_2_	44					
HCONH_2_	45					
?	46					
**Stage**		**I**	**II**		**III**	

## Data Availability

Not applicable

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
