# Peer review of "A Lizardite–HCN Interaction Leading the Increasing of Molecular Complexity in an Alkaline Hydrothermal Scenario: Implications for Origin of Life Studies"

_life, 2021, doi:10.3390/life11070661_

Round 1
Reviewer 1 Report
The present manuscript would be acceptable for publication, but it could be improved after minor modification with considering the following comments;
- The effect of serpentinite
As one of the conclusions, authors propose that serpentinite seems to inhibit the formation of bioorganic molecules. However, it is not clear whether the presence of serpentine inhibits (1) to form the bioorganic molecules, or (2) to detect those molecules by the analysis in the present study, because the argument in the section 3.4. XPS analysis (Lines 378-436) is not clear.
- Serpentine and its physicochemical features
``Serpentinite (Mg3Si2O5(OH)4) ''(Line 96) is inappropriate notation. Serpentinite is a name of the rock, composed mainly of serpentine, and serpentine is a general name of a subgroup of hydrous Mg-Fe phyllosilicate minerals. Serpentinite and serpentine should be distinguished and properly described, because natural serpentinites contain not only serpentine, but also montmorillonite, brucite, talc, magnetite, and others.
The serpentine subgroup minerals have three major polymorphs; antigorite, chrysotile and lizardite, with the same chemical formulae but different crystal structure. Chrysotile has a fiberous habit, and lizardite and antigorite have platy habit. Chrysotile and lizardite occur near the surface and break down at relatively low temperatures (< 400°C), whereas antigorite is stable at high temperatures upto ~600°C.
According to the XRD result, the serpentinite in this study is composed mainly of lizardite, so that a lizardite-related issue is discussed in the present paper. Lizardite is the most common serpentine polymorph, but it is not always certain that the result of this study with lizardite can be generalized to serpentine and serpentinite on the whole. Authors might infer chemical variaties (ratios of Mg/Fe2+, Fe3+/Fe2+, Al/Si -- these could be influenced by redox state and pH condition of hydrothermal system) and crystal structure (platy and fibrous habits, and related kinetics on the mineral surface) of the serpentine, in order to make clearer mechanism of the serpentinite-HCN interaction, and to emphasize a uniqueness of the present study.
- Additional minor comments
Figure 1A should be corrected accordingly. ``500mg serpentinite (> 1 cm)'' in the figure is inconsistent with ``fragments of mineral (< 1 cm)'' in text, Lines 99-100, and ``500 mg of clean serpentinite (particle size < 1 cm)'' in text, Line 112. And, authors have described that the serpentine sample was washed with KOH and HNO3 (and some of the sample was examined by using XRD), and then, 500 mg of the sample was added to HCN solution, in text, Lines 99-113. But in Figure 1, 500 mg serpentinite is directly put in the static system at ~ 100°C, without cleaning with KOH and HNO3. This point also should be correct.
Figures 1B and 1C are not very useful. Instead, might show enlarged images of a surface and/or a cross-section of fragments of the serpentinite covered by HCN-DTP (Phase III).
Author Response
Reviewer 1
The present manuscript would be acceptable for publication, but it could be improved after minor modification with considering the following comments;
- The effect of serpentinite
As one of the conclusions, authors propose that serpentinite seems to inhibit the formation of bioorganic molecules. However, it is not clear whether the presence of serpentine inhibits (1) to form the bioorganic molecules, or (2) to detect those molecules by the analysis in the present study, because the argument in the section 3.4. XPS analysis (Lines 378-436) is not clear.
First of all, we really appreciate your feedback. Thank you for your comment. In fact, what we noticed in our experiments is that the presence of the mineral affects the nature of the polymer, which was characterized by different techniques (i.e. thermal analysis, IR). After hydrolysis of the polymer, the amount and diversity of released organics was less than in the experiments without the mineral. The XPS analysis was performed to characterize the interactions among the polymer and the mineral. The fact that, it isIt is a low amount of organic molecule adsorbed on the mineral therefore, it is crucial the way we perform the XPS measurement in order to detect molecular signal. If we mix the sample making a pellet, the organic molecule get diluted in the bulk of the material. If we analyse only the surface of the mineral after making the reaction, XPS is able to emphasis the molecular adsorption and the nitrogen signal, fingerprint of the molecule, is detected. Nevertheless, we agree with the reviewer that from XPS results we cannot confirm that the mineral inhibit formation of the bioorganic molecules, we can confirm the adsorption or not of the molecules on the mineral surface.
-Serpentine and its physicochemical features
``Serpentinite (Mg3Si2O5(OH)4) ''(Line 96) is inappropriate notation. Serpentinite is a name of the rock, composed mainly of serpentine, and serpentine is a general name of a subgroup of hydrous Mg-Fe phyllosilicate minerals. Serpentinite and serpentine should be distinguished and properly described, because natural serpentinites contain not only serpentine, but also montmorillonite, brucite, talc, magnetite, and others.
The serpentine subgroup minerals have three major polymorphs: antigorite, chrysotile and lizardite, with the same chemical formulae but different crystal structure. Chrysotile has a fibrous habit, and lizardite and antigorite have platy habit. Chrysotile and lizardite occur near the surface and break down at relatively low temperatures (< 400°C), whereas antigorite is stable at high temperatures up to ~600°C.
Thank you for your comments, we totally agree with your observation. The name of the manuscript has been changed to:
“A lizardite-HCN interaction leading the increasing of molecular complexity in an alkaline hydrothermal scenario: Implications for the origin of life studies”
Also, in the Introduction a description of the relevance of the HV and serpentine was added, as follows:
“These cyanide species may have been scattered throughout some primitive environments such as the hydrothermal systems, either submarine and subaerial (Mukhin, 1974, 1976; Dowler & Ingmanson, 1979; Arrhenius et al., 1994; Keefe & Miller, 1996; Dzombak et al., 2006).
Hydrothermal environments, both sub-aerial and submarine, are considered as ideal system to allow chemical evolution on Earth (Martin et al. 2008; Colín-García, et al. 2016; Colín-García et al. 2018). It has been proposed that serpentine-hosted hydrothermal systems may support favorable conditions for prebiotic pathways, due to the coexistence of different geochemical variables on them (Schulte et al., 2006; Martin et al., 2008; Sleep et al., 2011; McCollom and Seewald, 2013).
Serpentinites are rocks formed mostly of serpentine-group minerals, derived from metamorphism of mafic-ultramafic rocks Serpentinites are rocks formed mostly of serpentine-group minerals, derived from metamorphism of mafic-ultramafic rocks which were abundant during Hadean-Archean (Schulte et al., 2006; Müntener, 2010; Arndt and Nisbet, 2012; Stüeken et al., 2013; Schrenk et al., 2013). In general, these hydrous magnesium silicates are formed after low-temperature (< 400 ºC) hydration of ferromagnesian or magnesian minerals (e.g., olivine, orthopyroxene) formed in basic and ultrabasic rocks (Evans et al., 2013).”
Also, the “2.1 Mineral/HCN samples” section was modified, some precision were made:
“Serpentinite (Mg3Si2O5(OH)4) was provided by Professor Fernando Ortega-Gutiérrez (Geology Institute, UNAM).”
“The clean mineral was characterized by XRD analysis. XRD analysis was performed to mineralogical characterization of the serpentinite sample. The XRD analysis showed that on the serpentine sample predominates the lizardite polymorph (Figure 2). In the sample predominates the lizardite polymorph. XRD spectra (Fig. 2) shows distinctive diffraction peaks of lizardite (91 %), antigorite (5%), and minor traces of magnetite and brucite (≈4%). Lizardite has the structural formula M3T2O5(OH)4, where M is mainly Mg and T is Si, although several common elements can be present in the structure such as Fe2+,3+, Al3+, Ni, Mn2+, Zn2+ (Rucklidge & Zussman, 1965; Mellini, 1981; Zheng & Wang, 2014). In this mineral, 1:1 flat layers of sheets of SiO4 tetrahedron and sheets of MgO2(OH)4 octahedron linked by hydrogen bonds. The most common polytypic is the stacking of three layers without any lateral shift (Carmignano et al., 2020).”
According to the XRD result, the serpentinite in this study is composed mainly of lizardite, so that a lizardite-related issue is discussed in the present paper. Lizardite is the most common serpentine polymorph, but it is not always certain that the result of this study with lizardite can be generalized to serpentine and serpentinite on the whole. Authors might infer chemical varieties (ratios of Mg/Fe2+, Fe3+/Fe2+, Al/Si -- these could be influenced by redox state and pH condition of hydrothermal system) and crystal structure (platy and fibrous habits, and related kinetics on the mineral surface) of the serpentine, in order to make clearer mechanism of the serpentinite-HCN interaction, and to emphasize a uniqueness of the present study.
We completely agree with the reviewer in the sense that chemical species (different ionic species of the same element) could influence the experiment. We are considering testing this in a future research.
- Additional minor comments
Figure 1A should be corrected accordingly. ``500mg serpentinite (> 1 cm)'' in the figure is inconsistent with ``fragments of mineral (< 1 cm)'' in text, Lines 99-100, and ``500 mg of clean serpentinite (particle size < 1 cm)'' in text, Line 112. And, authors have described that the serpentine sample was washed with KOH and HNO3 (and some of the sample was examined by using XRD), and then, 500 mg of the sample was added to HCN solution, in text, Lines 99-113. But in Figure 1, 500 mg serpentinite is directly put in the static system at ~ 100°C, without cleaning with KOH and HNO3. This point also should be correct.
The Figure was modified. An explanation of the previous treatment of the mineral was added and the hydrolysis procedure was also explained, both as notes.
Figures 1B and 1C are not very useful. Instead, might show enlarged images of a surface and/or a cross-section of fragments of the serpentinite covered by HCN-DTP (Phase III).
The Figures 1B and 1C were removed.
Reviewer 2 Report
To find origin of life scenarios it is highly important to consider plausible early earth environments especially how minerals interact with (in)organic molecules. Villafañe-Barajas et al. report HCN polymerisation under basic conditions in the presence of serpentinite mineral. They analyse the soluble, insoluble and remaining mineral (polymer coated) by various methods. Their main finding is that the presence of serpentinite reduces the molecular complexity of the identified products compared to similar HCN polymerisations under basic conditions. Especially, the formation of N-heterocycles was significantly reduced.
The authors provide good affords to characterise the reaction outcome but some methodological parameters need to be provided or are unclear.
1) Why did the authors choose to investigate serpentinite? Please elaborate on why serpentinite might be important in an origin of life scenario? How abundant is this mineral and where can it be found?
2) Please provide more details about how the HCN solution was prepared. Did the authors use HCN gas or basic HCN salts (such as KCN or NaCN). Does the counterion has an effect on the reaction outcome?
3) Was HCN polymerisation performed under inert atmosphere? Please clarify.
4) The serpentinite mineral was washed with KOH and HNO3, to remove organic molecules. However, can the authors show that indeed all organics are removed (e.g. GC-MS analysis after acid or basic treatment as performed for the HCN polymer). I think this is an important control experiment to make sure the reaction was not contaminated with pre-existing molecules.
5) Figure 8: What molecules (by GC-MS) do you find in the soluble phase (I) without acid or base treatment?
Minor comments:
Line 157: Please clarify how exactly the HCN-DTP control has been prepared. If this has been reported in another publication, please cite this here.
Line 412: What does really different mean here? Please elaborate what the main differences are.
Line 445: According to the methods part acid hydrolysis has been performed at 110C (line 149). In line 445 it says 100C. Also, acid hydrolysis is much harsher (24h, 6M) compared to base hydrolysis (6h, 0.1M) could you elaborate on the reason for this?
Author Response
Reviewer 2
To find origin of life scenarios it is highly important to consider plausible early earth environments especially how minerals interact with (in)organic molecules. Villafañe-Barajas et al. report HCN polymerization under basic conditions in the presence of serpentinite mineral. They analyse the soluble, insoluble and remaining mineral (polymer coated) by various methods. Their main finding is that the presence of serpentinite reduces the molecular complexity of the identified products compared to similar HCN polymerizations under basic conditions. Especially, the formation of N-heterocycles was significantly reduced.
The authors provide good affords to characterize the reaction outcome but some methodological parameters need to be provided or are unclear.
1) Why did the authors choose to investigate serpentinite? Please elaborate on why serpentinite might be important in an origin of life scenario? How abundant is this mineral and where can it be found?
Thank you for all your comments.
We consider important to explain what you asked. In the Introduction section a paragraph related to the relevance of serpentinite was added.
“Serpentinites are rocks formed mostly of serpentine-group minerals, derived from metamorphism of mafic-ultramafic rocks which were abundant during Hadean-Archean (Schulte et al., 2006; Müntener, 2010; Arndt and Nisbet, 2012; Stüeken et al., 2013; Schrenk et al., 2013). In general, these hydrous magnesium silicates are formed after low-temperature (< 400 ºC) hydration of ferromagnesian or magnesian minerals (e.g., olivine, orthopyroxene) formed in basic and ultrabasic rocks (Evans et al., 2013).”
2) Please provide more details about how the HCN solution was prepared. Did the authors use HCN gas or basic HCN salts (such as KCN or NaCN). Does the counterion has an effect on the reaction outcome?
The procedure was fully described on text.
“The HCN-DTP in presence of serpentinite was synthesized as follows. HCN solution was produced in situ by the reaction between KCN and H2SO4 under argon atmosphere (for further details, please see Villafañe-Barajas et al., 2020). Once the desired concentration was reached (0.15 mol L-1), the pH of the HCN solution was adjusted (pH > 10) with KOH solution (0.1 mol L-1) to favor the availability of CN- and the formation of HCN-polymer. Finally, aliquots of HCN solution (0.15 mol L-1, 5 mL) were prepared with 500 mg of the previously cleaned serpentinite in glass tubes and heated in a static system at 100 °C for 50 h. The selected temperature is consistent with the one found in the surroundings of alkaline hydrothermal environments. After treatment, three phases could be distinguished in the sample: I) supernatant (yellow soluble part), II) HCN-DTP (black polymer that was not adhered to mineral surface) and III) Mineral + HCN-DTP (serpentinite covered by polymer). The three phases were analyzed by different analytical techniques (for more details, see Figure 1).”
3) Was HCN polymerization performed under inert atmosphere? Please clarify.
The polymer was synthesized under an argon atmosphere. A line related to this was added.
“HCN solution was produced in situ by the reaction between KCN and H2SO4 under argon atmosphere as previous reports (for further details, please see Villafañe-Barajas et al., 2020).”
4) The serpentinite mineral was washed with KOH and HNO3, to remove organic molecules. However, can the authors show that indeed all organics are removed (e.g. GC-MS analysis after acid or basic treatment as performed for the HCN polymer). I think this is an important control experiment to make sure the reaction was not contaminated with pre-existing molecules.
We agree with you, that it is really important to have a good control in all the experiments. A thermal analysis-mass spectroscopy (TG-MS) of the control sample (the mineral alone) was performed, and we found no organic carbon signals on it. So, we could be sure that there is not contamination in the mineral sample as it is now explained in the main text.
“To warranty that the mineral sample was not contaminated, a mass spectroscopy thermal analysis was carried out for the cleaned serpentinite sample. Peaks related to OH- and H2O (at 619 and 696 ºC) were detected, this corroborated that all organic material was removed from serpentinite”. (Figure not shown in the text).
5) Figure 8: What molecules (by GC-MS) do you find in the soluble phase (I) without acid or base treatment?
The raw soluble phase was not analyzed in these experiments. In all cases, we carried out a hydrolysis treatment of samples. The main objective was to understand the differences on the HCN-derived polymer and to evaluate the role of the mineral. Of course, we are interested in understanding as much as we can the system. In the future we also want to analyze the soluble phase.
Minor comments:
Line 157: Please clarify how exactly the HCN-DTP control has been prepared. If this has been reported in another publication, please cite this here.
Thank you, we already have detailed the procedure.
Line 412: What does really different mean here? Please elaborate what the main differences are.
We explain, along the text, what we meant:
“The assignation of these components is really different to others coatings obtained from AMN at room temperature (although N 1s signal is significant, the authors performed deep analysis to C 1s spectra and the binding energies were referenced to aliphatic hydrocarbons peak at 285.0 eV; Thissen et al., 2015) or insoluble cyanide polymers (397.6 eV to imine and/or heterocyclic groups and 398.7 eV to amides; Ruiz‐Bermejo et al., 2012).”
Line 445: According to the methods part acid hydrolysis has been performed at 110C (line 149). In line 445 it says 100C. Also, acid hydrolysis is much harsher (24h, 6M) compared to base hydrolysis (6h, 0.1M) could you elaborate on the reason for this?
It lacked precision in the description. We have already corrected it as follows:
“we carried out a hydrolysis procedure (both alkaline and acid hydrolysis at 100 – 110 ºC, respectively) for each phase…”
We have chosen these hydrolysis conditions following the method of Ferris et al. 1974 (Ferris, J.P.; Wos, J.D.; Nooner, D.W.; Oró, J. Chemical evolution: XXI. The Amino Acids Released on Hydrolysis of HCN Oligomers. J. Mol. Evol. 1974, 3, 225–231). This is considered the reference method for HCN-polymers (oligomers) analysis. In addition, this allowed us to compare our results with other works.

Reviewer 3 Report
HCN is indeed an important molecule for prebiotic evolution. This study builds on the previous research done by the same group, and this time, the effect of a representative mineral, serpentine, was studied in terms of polymerization of HCN and other reactivity. Integrated analytic methods, including thermal (TG-DSC), chemical (FT-IR), and compositional (MS), and surface (XPS) analyses, were conducted, with comprehensive discussions. Overall speaking, this is a very comprehensive study. I only have one major concern as below:
Line 189. It was suggested that "The serpentinite increases the hydrolysis /oxidation processes in the HCN polymerization leading to highly oxidized products." First of all, how does oxidation happen? On the early Earth, an anaerobic environment is more realistic. Unfortunately, the experiments seem to be conducted without atmospheric control. A careful degassing using either N2 or Ar during the experiment is needed to simulate the early Earth environment.
Author Response
Reviewer 3
HCN is indeed an important molecule for prebiotic evolution. This study builds on the previous research done by the same group, and this time, the effect of a representative mineral, serpentine, was studied in terms of polymerization of HCN and other reactivity. Integrated analytic methods, including thermal (TG-DSC), chemical (FT-IR), and compositional (MS), and surface (XPS) analyses, were conducted, with comprehensive discussions. Overall speaking, this is a very comprehensive study. I only have one major concern as below:
Line 189. It was suggested that "The serpentinite increases the hydrolysis /oxidation processes in the HCN polymerization leading to highly oxidized products." First of all, how does oxidation happen? On the early Earth, an anaerobic environment is more realistic. Unfortunately, the experiments seem to be conducted without atmospheric control. A careful degassing using either N2 or Ar during the experiment is needed to simulate the early Earth environment.
Thank you very much for your time and comments.
You are right in your observation; it was a great omission of our part to mention that the polymer was synthesized under an argon atmosphere. This was clarified in the procedure. So, we had anaerobic experiments. The word oxidation was deleted and the sentence was corrected as follows:
“Therefore, it seems that the serpentinite increases the hydrolysis/oxidation processes in the HCN polymerization, resulting into highly oxidized products”.

Round 2
Reviewer 2 Report
The authors responded to all points mentioned in the first round of review. I am in favour of publication.
Reviewer 3 Report
The authors have replied to my concern. Now I can understand that their experiments were conducted under anaerobic conditions.